# Antibody Response of BNT162b2 and CoronaVac Platforms in Recovered Individuals Previously Infected by COVID-19 against SARS-CoV-2 Wild Type and Delta Variant

**DOI:** 10.3390/vaccines9121442

**Published:** 2021-12-07

**Authors:** Ruiqi Zhang, Ka-Wa Khong, Ka-Yi Leung, Danlei Liu, Yujing Fan, Lu Lu, Pui-Chun Chan, Linlei Chen, Kelvin Kai-Wang To, Honglin Chen, Kwok-Yung Yuen, Kwok-Hung Chan, Ivan Fan-Ngai Hung

**Affiliations:** 1Department of Medicine, Li Ka Shing Faculty of Medicine, University of Hong Kong, Hong Kong, China; zhangrq@hku.hk (R.Z.); kwkhong@connect.hku.hk (K.-W.K.); danlei6@connect.hku.hk (D.L.); jyjfan@connect.hku.hk (Y.F.); 2Department of Microbiology, Li Ka Shing Faculty of Medicine, University of Hong Kong, Hong Kong, China; joy2u@connect.hku.hk (K.-Y.L.); u3003963@connect.hku.hk (L.L.); bcpc@hku.hk (P.-C.C.); u3006707@connect.hku.hk (L.C.); kelvinto@hku.hk (K.K.-W.T.); hlchen@hku.hk (H.C.); kyyuen@hku.hk (K.-Y.Y.); chankh2@hku.hk (K.-H.C.); 3State Key Laboratory for Emerging Infectious Diseases, Li Ka Shing Faculty of Medicine, University of Hong Kong, Hong Kong, China; 4Carol Yu Centre for Infection, Li Ka Shing Faculty of Medicine, University of Hong Kong, Hong Kong, China

**Keywords:** COVID-19, vaccines, Delta variants

## Abstract

Vaccinating recovered patients previously infected by COVID-19 with mRNA vaccines to boost their immune response against wild-type viruses (WT), we aimed to investigate whether vaccine platform and time of vaccination affect immunogenicity against the SARS-CoV-2 WT and Delta variant (DV). Convalescent patients infected by COVID-19 were recruited and received one booster dose of the BNT162b2 (PC-B) or CoronaVac (PC-C) vaccines, while SARS-CoV-2 naïve subjects received two doses of the BNT162b2 (CN-B) or CoronaVac (CN-C) vaccines. The neutralizing antibody in sera against the WT and DV was determined with live virus neutralization assay (vMN). The vMN geometric mean titre (GMT) against WT in recovered individuals previously infected by COVID-19 reduced significantly from 60.0 (95% confidence interval (CI), 46.5–77.4) to 33.9 (95% CI, 26.3–43.7) at 6 months post recovery. In the PC-B group, the BNT162b2 vaccine enhanced antibody response against WT and DV, with 22.3-fold and 20.4-fold increases, respectively. The PC-C group also showed 1.8-fold and 2.2-fold increases for WT and DV, respectively, after receiving the CoronaVac vaccine. There was a 10.6-fold increase in GMT in the CN-B group and a 1.3-fold increase in the CN-C group against DV after full vaccination. In both the PC-B and PC-C groups, there was no difference between GMT against WT and DV after vaccination. Subjects in the CN-B and CN-C groups showed inferior GMT against DV compared with GMT against WT after vaccination. In this study, one booster shot effectively enhanced the pre-existing neutralizing activity against WT and DV in recovered subjects.

## 1. Introduction

Since its outbreak in 2019, coronavirus disease 2019 (COVID-19) has become an unprecedented pandemic and has cost millions of lives. COVID-19 is caused by severe acute respiratory syndrome coronavirus 2 (SARS-CoV-2), which is a type of enveloped and positive-stranded RNA virus [1]. The receptor-binding domain (RBD) of spike protein, which is the membrane protein of SARS-CoV-2, can bind to angiotensin-converting enzyme 2 (ACE2) on the surface of host cells [2]. The virus then enters the cell and completes viral replication. New virions can transmit among the human population through aerosol, droplets and fomites [3]. In response to the COVID-19 pandemic, several diagnosis techniques for COVID-19 have been developed. There are three major methods for diagnosis: polymerase chain reaction (PCR)-based molecular tests, rapid antigen and antibody tests, and immune enzymatic serological tests [4]. In addition, viral culture and imaging techniques are applied in COVID-19 diagnosis [4]. Although these techniques can help us screen new COVID-19 cases quickly and efficiently to tackle the pandemic, vaccines are still the primary method used to control COVID-19. COVID-19 vaccines approved for clinical use include mRNA vaccines (BNT162b2), viral vector-based vaccines (ChAdOx1), inactivated virus vaccines (CoronaVac), and subunit protein vaccines (NVX-CoV2373) [5]. Vaccines can induce immune response against SARS-CoV-2, which can also be elicited by virus infection, to protect the host [6]. Some studies suggested that, after SARS-CoV-2 infection, antibody levels in the serum of a recovered patient can persist for at least 3 months [6,7]. However, the protection against repeated infection is estimated to be only around 80% and the presence of virus mutation has been shown to result in >10-fold reduction in virus neutralization by convalescent plasma from patients infected by COVID-19, which greatly increases the risk of reinfection [8,9]. In Hong Kong, a case of re-infection by phylogenetically distinct SARS-CoV-2 strains was detected and confirmed [10]. Therefore, with the increasing prevalence of B.1.617.2 (Delta variant, DV) and a possible fading antibody level, risk of re-infection is a big obstacle to the resolution of the pandemic.

It was previously demonstrated that an additional dose of mRNA vaccine given to recovered patients previously infected by COVID-19 enhances immune responses and protection against virus variants [11,12]. In addition, mRNA vaccines were demonstrated to be protective against DV infection in healthy individuals [13].

Different vaccine platforms have been adopted across the globe, and in Hong Kong, both the BNT162b2 and CoronaVac vaccines are available [5,14]. The knowledge of whether vaccine platforms affect the boosting effect is an important consideration for policy making. In the present study, we aim to evaluate the persistence of antibody level in recovered patients previously infected by COVID-19, the boosting effect on the antibody levels of these patients with different vaccine platforms, whether such vaccinations protect them from DV, and the most appropriate timing for administering the vaccine.

## 2. Materials and Methods

### 2.1. Study Design and Participants

This is a prospective cohort study performed in the Hong Kong West Cluster Hospitals under the Hospital Authority in Hong Kong. We followed up with patients who have recovered from a COVID-19 infection by taking their blood after discharge and at 6 weeks, 12 weeks, 6 months and 1 year post recovery. We then compared their neutralizing antibody responses after single-dose BNT162b2 or CoronaVac vaccination against those who were SARS-CoV-2 naïve and received two doses of either the BNT or CoronaVac vaccination at the same time after their first dose.

Recruited participants were vaccinated according to their preference. SARS-CoV-2 naïve individuals aged above eighteen were recruited and administered two doses of the BNT162b2 or CoronaVac vaccines. Their blood samples were collected at the following timepoints: before vaccination (baseline), 21 days (BNT162b2) or 28 days (CoronaVac) after the first dose, and 56 days after the primer dose. In addition, we recruited both recovered patients previously infected by COVID-19 from the first part of the study and other recovered patients previously infected by COVID-19 who were not recruited in the first part of the study. Recruited participants were given one dose of a vaccine and had their blood sampled at the following timepoints: baseline and 28 days after their first dose. The study was approved by the institutional review board of the University of Hong Kong and Hospital Authority (UW 21-214).

### 2.2. Procedure

The nurse administered the vaccines as an intramuscular injection according to the participant’s choice. Recruited participants were then assigned to 4 groups depending on the vaccine they chose: recovered patients previously infected by COVID-19 who chose BNT162b2 were given a single dose of IM BNT162b2 (0.3 mL); recovered patients previously infected by COVID-19 who chose CoronaVac were given a single dose of IM CoronaVac (0.5 mL); SARS-CoV-2 naïve participants who chose BNT162b2 were given two doses of IM BNT162b2 (0.3 mL), 21 days apart; and SARS-CoV-2 naïve participants who chose CoronaVac were given two doses of IM CoronaVac (0.5 mL), 28 days apart (Figure 1).

Blood was taken from the participants at the timepoints mentioned above for use in an antibody assay. As described earlier, live virus microneutralization assay (vMN) was performed in the Biosafety Level 3 facility at HKU to determine the neutralizing antibody in sera [15]. Serial 2-fold dilutions of serum starting from 1:10 were incubated with 100 median tissue culture infectious doses (TCID50) of SARS-CoV-2 HKU-001a (wild type, GenBank accession number MT230904) strain (WT) [16] or DV for 1.5 h at 37 °C. Then, a serum–virus mixture was added to VeroE6/TMPRSS2 cells (JCRB Cell Bank Catalogue no. JCRB1819) on 96-well plates [17]. After 72 h of incubation at 37 °C and 5% CO_2_, the cytopathic effect (CPE) was examined and the antibody titre was determined by the highest dilution with 50% inhibition of CPE.

### 2.3. Outcome

The primary endpoint of this study was the vMN geometric mean titre (GMT) against WT and DV. The secondary endpoints were GMT fold increase and safety. In terms of safety, severe adverse events (SAE) defined as vaccine-related death, or disabling or life-threatening conditions were recorded.

### 2.4. Statistical Analysis

A statistical inference of normally distributed continuous variables was performed using *t*-tests, including demographic parameters (age), GMT, and GMT fold increase compared with two-tailed *t*-tests. The comorbidities and severity of COVID-19 were analyzed with χ^2^ test. When *p* < 0.05, the result was statistically significant. SPSS Statistics 27.0 was used for statistical computation.

## 3. Results

### 3.1. Neutralizing Activity in Individuals after Recovery from COVID-19

Between April 2020 and July 2021, 79 recovered patients previously infected by COVID-19 were recruited to the study. All of the participants were recovered patients previously infected by COVID-19 and were not vaccinated. Among these patients, there were 9 severe and 70 mild COVID-19 cases (Table 1). No difference in the percentage of severe COVID-19 cases was found between the PC-B and PC-C groups (*p* = 0.38). We observed a persistent reduction in serum neutralizing antibodies level against SARS-CoV-2 after discharge (*p* = 0.0027). The vMN GMT at baseline is 60.0 (95% confidence interval (CI), 46.5–77.4), which falls to 42.3 (95% CI, 33.7–53.0), 35.7 (95% CI, 27.4–46.5), 33.9 (95% CI, 26.3–43.7), and 18.3 (95% CI, 9.1–36.9) at 6 weeks, 12 weeks, 6 months, 1 year post recovery, respectively, demonstrating a persistent reduction in the level of serum neutralizing antibodies (Figure 2).

### 3.2. Comparison of the Immunogenicity of COVID-19 Vaccines in Recovered Patients Previously Infected by COVID-19 and SARS-CoV-2 Naïve Individuals

Between May and August 2021, we recruited 149 recovered patients previously infected by COVID-19 and healthy individuals to evaluate the protection provided by the BNT162b2 vaccine and the CoronaVac vaccine. Recruited participants were assigned to four groups depending on the vaccine they received: (use median age for all) recovered patients previously infected by COVID-19 received a booster dose of the BNT162b2 (PC-B, *n* = 54, median age = 50.5 years) or CoronaVac (PC-C, *n* = 25, median age = 55 years) vaccines, and SARS-CoV-2 naïve subjects received two doses of the BNT162b2 (CN-B, *n* = 34, mean median age = 50 years) or CoronaVac (CN-C, *n* = 36, median age = 61 years) vaccines (Table 1). There was no statistically significant difference in age between PC-B and CN-B (*p* = 0.169), and between PC-C and CN-C (*p* = 0.057). In terms of comorbidities, there was also no significant difference between PC-B and CN-B (*p* = 0.935), and between PC-C and CN-C (*p* = 0.804). At baseline, recovered patients recruited to the PC-B and PC-C groups had similar GMT against WT (PC-B, 32.2 (95% CI, 24.8–41.6); PC-C, 32.9 (95% CI, 24.7–44.0)) (*p* = 0.441) and DV (PC-B, 37.5 (95% CI, 28.4–49.6); PC-C, 32.9 (95% CI, 23.3–46.6)) (*p* = 0.510). In addition, SARS-CoV-2 naïve individuals recruited to the CN-B and CN-C groups had no GMT against WT and DV (Table 2).

When evaluating the immunogenicity of vaccination within each group, a booster shot of the BNT162b2 vaccine given to recovered patients previously infected by COVID-19 (PC-B) greatly enhanced the GMT against both WT (GMT fold increase, 22.3 (95% CI, 16.2–30.8)) and DV (GMT fold increase, 20.4 (95% CI, 14.8–28.1)) (Table 2). For subjects from the CN-B group, a booster dose was needed to induce a high level of neutralizing antibodies against WT and DV (Table 2). The subjects of PC-B showed a significantly higher GMT fold increase against the Delta variant than those receiving two doses of vaccine from CN-B (*p* = 0.013) (Figure 3c). Similarly, a GMT fold increase against DV in the PC-C group after receiving a CoronaVac vaccine booster was 2.2 (95% CI, 1.4–3.3) and significantly higher than that in the CN-C group receving a post booster dose of CoronaVac (1.2 (95% CI, 1.0–1.4), *p* = 0.029) (Figure 3c). Furthermore, SARS-CoV-2-naïve subjects showed a significantly lower level of antibodies against DV than against WT after receiving one dose of BNT162b2 (*p* = 0.003) or two doses of CoronaVac (*p* = 0.001) (Figure 4c,d). In contrast, after vaccination, recovered individuals previously infected by COVID-19 showed similar levels of neutralizing antibodies against the WT and DV strains in both the PC-B (*p* = 0.554) and PC-C groups (*p* = 0.601) (Figure 4a,b). In this study, no SAE was reported after vaccination.

### 3.3. Effect of Time from Recovery on Immunogenicity of COVID-19 Vaccines in Recovered Individuals Previously Infected by COVID-19

We also evaluated the effect of time from discharge on immunogenicity of different vaccines in recovered patients previously infected by COVID-19. There was no significant difference in GMT between participants who received one booster shot within 6 months of discharge and after 6 months of discharge, regardless of the vaccine platform they chose (Table 3).

## 4. Discussion

COVID-19 is a persisting pandemic caused by infections of SARS-CoV-2. With the emergence of variants, protection against SARS-CoV-2 re-infection is a major concern for both recovered patients previously infected by COVID-19 and SARS-CoV-2 naïve individuals who are vaccinated. Our group previously identified a case of SARS-CoV-2 re-infection by a phylogenetically distinct SARS-CoV-2 variant, and further analysis revealed that serum neutralizing antibodies were not detected until day 3 after hospitalization, contributing to re-infection [10,18]. From the current study, we observed a persisting decline in the level of serum neutralizing antibodies in recovered patients previously infected by COVID-19; therefore, interventions should be conducted to boost the antibodies level in these patients to maintain protection against SARS-CoV-2 re-infection.

Previous studies have demonstrated that, by giving participants with a previous infection an additional dose of the mRNA vaccine, there is enhanced humoral immune responses as well as increased protection against the B.1.1.7 and B.1.351 variants [11,12]. Interestingly, mRNA vaccines were demonstrated to be effective at inducing antibody response against the SARS-CoV-2 variants and to prevent symptomatic and severe COVID-19 associated with DV infection in healthy individuals who received two doses [13,19]. From our study it is evident that one booster dose, in particular, the mRNA vaccine, induced 22.3-fold (95% CI, 16.2–30.8) and 20.4-fold (95% CI, 14.8–28.1) enhancements of vMN GMT against WT and DV, respectively. Furthermore, in comparison with SARS-CoV-2 naïve individuals (CN-B group), recovered patients from the PC-B group had much more potent immune responses, with a higher GMT and GMT fold increase. One plausible explanation is immune memory, especially memory B cells, in recovered patients [20]. Upon re-exposure to SARS-CoV-2, memory B cells against SARS-CoV-2 can rapidly proliferate and differentiate into plasma cells, which generate antibodies to prevent reinfection [20]. Dan et al. found that the number of memory B cells against SARS-CoV-2 spike proteins increase between 1 month and 8 months post infection [21]. Another explanation could be due to a longer germinal centre reaction in recovered patients previously infected by COVID-19 compared with healthy individuals; knowledge from a study of HIV vaccination in rhesus macaque reveals that a successful immune response depends on the development of neutralizing antibodies and is associated with germinal centre responses. Interestingly, a SARS-CoV-2 mRNA vaccine can induce a persistent human germinal centre response for at least 12 weeks [22,23]. In our study, recovered patients previously infected by COVID-19 received their booster dose and their blood was sampled at least a few months after they were discharged, while healthy recipients had their blood sampled 3–8 weeks following the first dose of vaccine. It is possible, then, that recovered patients previously infected by COVID-19 underwent a more complete series of germinal centre reactions, resulting in a better neutralizing antibody titre.

Furthermore, we also questioned whether a similar effect can be achieved by inactivated virus vaccines (CoronaVac). Previously, test-negative case–control studies have suggested that both mRNA vaccines and inactivated vaccines are effective against DV, but serological data are lacking [24,25]. Our study supplements the knowledge gap and demonstrates that inactivated vaccines, although not as potent as mRNA vaccines, also enhance the neutralizing antibody level against both WT and DV in recovered individuals previously infected by COVID-19 (Table 2).

In addition to serum neutralization antibody level, whether these antibodies, induced by vaccines or an infection, are protective against DV is a concern. Several studies have demonstrated that DV shows reduced sensitivity to neutralizing antibodies in sera from healthy individuals after vaccination and causes a lower effectiveness of the vaccine [26,27,28]. In this study, both the BNT162b2 and CoronaVac vaccines also elicited a reduction in antibody response against DV compared with WT in non-infection subjects (BNT162b2, post first dose, *p* = 0.003; CoronaVac, post second dose, *p* = 0.001) (Figure 4c,d). However, no difference between neutralizing activity against DV and WT strains was observed in recovered individuals previously infected by COVID-19 after receiving a booster. Lucas et al. also reported that the neutralizing activity against the DV strain decreases 1.5-fold compared with the control virus strain (lineage A) in vaccinees with previous infection but reduces 6.9-fold in healthy vaccinees [26]. The observation may be explained by the evolution of viruses during infection. In the course of infection, viruses may undergo mutation during replication, and patients can be exposed to multiple antigens within the infection period. By sequencing serial respiratory specimens from patients infected by COVID-19, our department has identified mutations in spike proteins within the infection period of a patient [29]. Compared with SARS-CoV-2 naïve patients who are only presented with a single epitope, the diversity in neutralizing antibody epitopes during infection can lead to the development of polyclonal antibodies and the selection of pan-neutralizing antibodies, inducing a robust protection against both WT and DV after infection.

The timing of the vaccination is important. In Hong Kong, it is recommended by the Department of Health that recovered patients previously infected by COVID-19 should wait for at least 90 days after discharge from previous infection before they receive a BNT162b2 vaccine, while recovered patients previously infected by COVID-19 who wish to receive the CoronaVac vaccine should receive the vaccine at least 180 days after discharge from a previous infection [30]. The rationale behind this recommendation could be the effect of pre-existing antibodies on vaccine efficacy. A recent review on the effect of pre-existing antibodies on vaccine responses suggested that pre-existing immunity can reduce the efficacy of both inactivated and live-attenuated vaccines for influenza, which is also a respiratory RNA virus [31]. Since we observed a persisting reduction in serum neutralizing antibody level up to 6 months post discharge, we divided our participants into those who received the vaccine before 6 months post discharge and those who received the vaccine after 6 months post discharge. Interestingly, our results show that there was no significant difference between vaccination before or after 6 months, regardless of the vaccine platform. As there was significant reduction in vMN GMT at 12 weeks and 6 months after discharge (Figure 1), patiented infected by COVID-19 may need to receive a booster between 3 and 6 months after recovery. For the third booster dose in vaccinees, two clinical studies report a reduced effectiveness of the BNT162b2 vaccine against SARS-CoV-2 at 6 months after their first dose [32,33]. Thus, vaccinees should receive their third dose of a COVID-19 vaccine 5 months after full vaccination.

In the laboratory, an immune response induced by SARS-CoV-2 infection and vaccines can be evaluated by testing cytokine expression, antibody levels, and the number of T lymphocytes [34,35,36]. The anti-SARS-CoV-2 antibody can be detected via vMN, ELISA, and immunofluorescence tests (IF) [15,37]. ELISA and IF detect antibodies binding to SARS-CoV-2. These antibodies contain neutralizing antibodies that protect the host by inhibiting viral replication and non-neutralizing antibodies. Recently, a surrogate neutralizing antibody (sNAb) test based on ELISA was developed [34]. In this technique, RBD of the SARS-CoV-2 spike protein was used to determine neutralizing antibodies that bind to RBD. Compared with the sNAb test, vMN used in the study can detect neutralizing antibodies including not only the anti-RBD antibody but also the anti-other epitope neutralizing antibody, for instance, antibodies against the N-terminal domain (NTD) of spike [15,38]. Thus, the results from vMN can be used to comprehensively evaluate the neutralizing activity induced by vaccine and infection.

One limitation is our small sample size, and further investigation with more participants will allow us to define a better timepoint of post-COVID vaccination. The other limitation of this study is the lack of clinical protection data and long-term immunogenicity results; current knowledge suggests that antibodies induced by mRNA vaccine can persist up to 6 months after the second dose [39]. Long-term follow-up is required to evaluate the changes in antibody level in patients infected by COVID-19 who received one dose of the vaccine; this information is important to decide whether an annual booster dose is needed. Cellular immune response should also be assessed in future studies.

## 5. Conclusions

Our study found that one BNT162b2 or CoronaVac vaccine booster dose could enhance the pre-existing antibody response in recovered patients against WT and DV effectively. Furthermore, our study demonstrated that, for SARS-CoV-2 naïve individuals, two doses of BNT162b2 protects against DV, but for individuals who received two doses of CoronaVac, protection against DV is minimal; therefore, an additional mRNA booster dose should be considered to enhance their neutralizing antibody response against SARS-CoV-2 variants. This study could provide useful information about the time of vaccination in recovered individuals previously infected by COVID-19.

## Figures and Tables

**Figure 1 vaccines-09-01442-f001:**
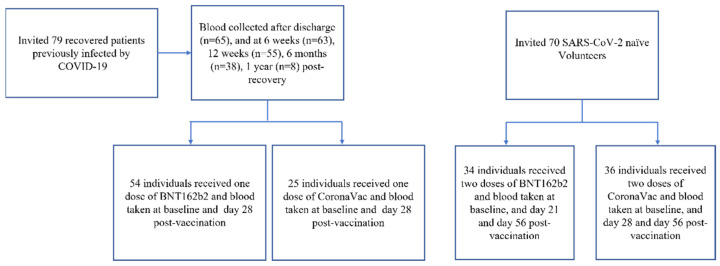
Trial profile.

**Figure 2 vaccines-09-01442-f002:**
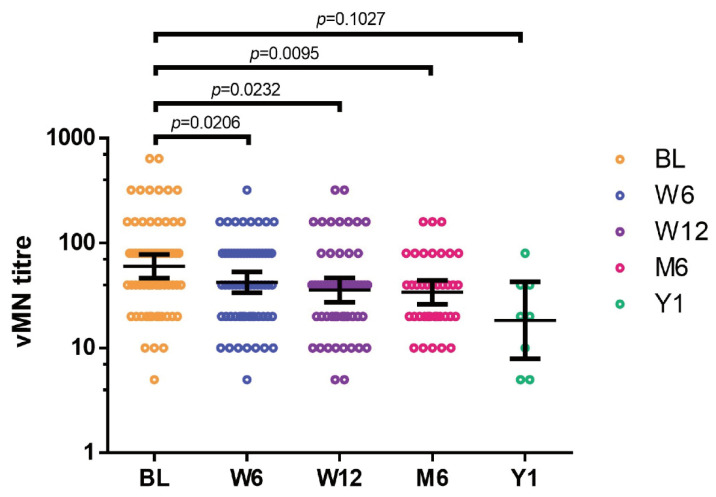
Change in neutralizing activity in participants after recovering from COVID-19. Convalescent patients infected by COVID-19 aged above 18 years were recruited, and blood samples were taken after discharge and at 6 weeks, 12 weeks, 6 months, and 1 year post recovery. vMN was used to determine the neutralizing antibody in sera. BL: baseline; W6: 6 weeks post recovery; W12: 12 weeks post recovery; M6: 6 months post recovery; Y1: 1 year post recovery; *p* < 0.05, statistically significant difference. Each dot represents an individual serum sample, and the error bar represents the 95% confidential interval (CI).

**Figure 3 vaccines-09-01442-f003:**
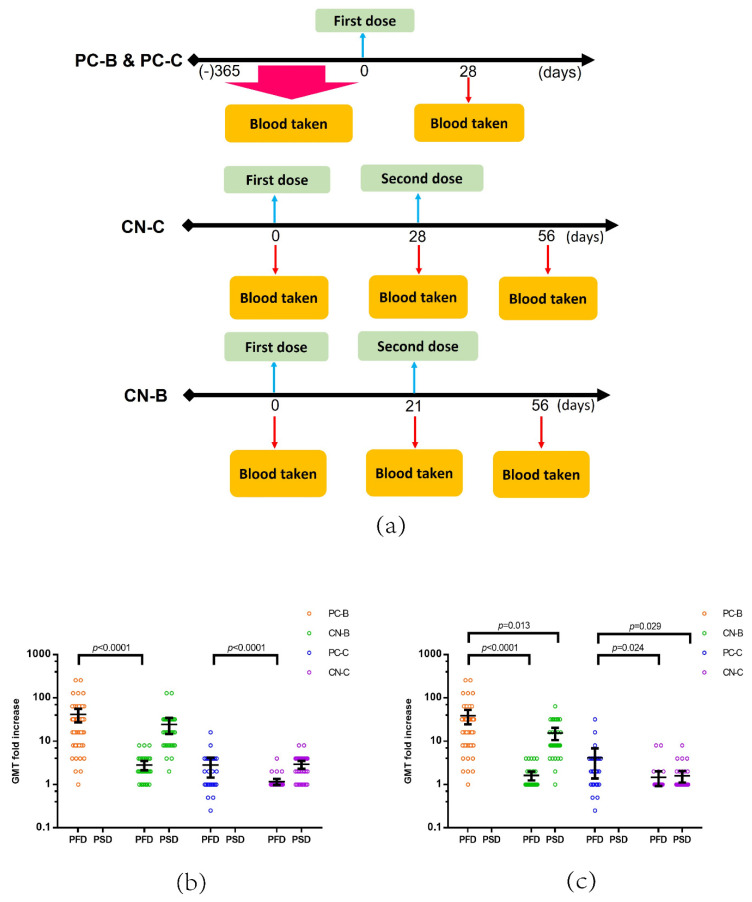
GMT fold change after vaccination. (**a**) Recovered individuals previously infected by COVID-19 received a booster dose of BNT162b2 (PC-B) or CoronaVac (PC-C) and had their blood taken at baseline and at day 28 post vaccination. SARS-CoV-2 naïve participants received two doses of BNT 162b2 (CN-B) or CoronaVac (CN-C) and has their blood collected at baseline, on day 21 (CN-B) and day 28 (CN-C), and on day 56 after their first dose. Neutralizing antibody titre in serum was determined with vMN, and GMT change against the wild type (**b**) and Delta variant (**c**) was determined by comparing the antibody titre after vaccination with the antibody titre at baseline. PFD: post first dose; PSD: post second dose.

**Figure 4 vaccines-09-01442-f004:**
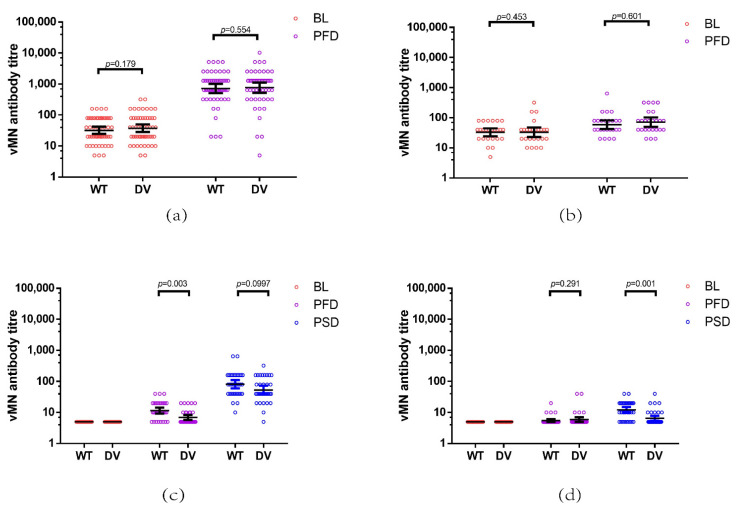
Comparison of neutralizing antibody titre against wild type and Delta variant after vaccination. Recovered individuals previously infected by COVID-19 received a booster of BNT162b2 (PC-B) or CoronaVac vaccines (PC-C) and had their blood taken at baseline and day 28 post vaccination. SARS-CoV-2 naïve participants received two doses of BNT 162b2 (CN-B) or CoronaVac (CN-C) and had their blood collected at baseline, on day 21 (CN-B) or day 28 (CN-C), and on day 56 after their first dose. vMN was used to determine neutralizing antibody titre in sera from (**a**) PC-B, (**b**) PC-C, (**c**) CN-B, and (**d**) CN-C. WT: wild type; DV: Delta variant; BL: baseline; PFD: post first dose; PSD: post second dose.

**Table 1 vaccines-09-01442-t001:** Baseline characteristics of subjects receiving COVID-19 vaccines.

	PC-B (*n* = 54)	CN-B (*n* = 34)	PC-C (*n* = 25)	CN-C (*n* = 36)
Age (years)	50.5 (23–80)	50 (18–68)	55 (21–73)	61 (20–76)
Female	19 (35.2%)	22 (64.7%)	7 (28%)	22 (61.1%)
Male	35 (64.8%)	12 (35.3%)	18 (72%)	14 (38.9%)
Severity of COVID-19				
Severe	5 (9.3%)	N/A	4 (16%)	N/A
Mild	49 (90.7%)	N/A	21 (84%)	N/A
Comorbidities	13 (24.1%)	8 (23.5%)	11 (44%)	17 (47.2%)

Data are median age (range) or *n* (%); PC-B: recovered individuals previously infected by COVID-19 receiving BNT162b2; PC-C: recovered individuals previously infected by COVID-19 receiving CoronaVac; CN-B: SARS-CoV-2 naïve individuals receiving BNT162b2; CN-C: SARS-CoV-2 naïve individuals receiving CoronaVac. Comorbidities: hypertension (HT), ischemic heart diseases (IHD), diabetes mellitus (DM), stroke, chronic heart failure (CHF), malignancy, asthma, chronic obstructive pulmonary disease (COPD), and thyroid diseases. N/A: not applicable.

**Table 2 vaccines-09-01442-t002:** Immunogenicity of COVID-19 vaccines in recovered subjects and SARS-CoV-2-naïve subjects.

	PC-B (*n* = 54)	CN-B (*n* = 34)	PC-C (*n* = 25)	CN-C (*n* = 36)
**Wild type**				
Baseline				
GMT	32.2 (24.8–41.6)	5 (5–5)	32.9 (24.7–44.0)	5 (5–5)
Post primer dose				
GMT		11.5 (14.3–9.3)		5.5 (5.0–6.1)
GMT fold increase value		2.3 (1.9–2.9)		1.1 (1.0–1.2)
Post booster dose				
GMT	718.4 (513.2–1005.7)	81.6 (60.4–110.3)	59.0 (43.1–80.8)	12.1 (9.9–14.9)
GMT fold increase value	22.3 (16.2–30.8)	16.3 (12.1–22.1)	1.8 (1.2–2.6)	2.4 (2.0–3.0)
**Delta variant**				
Baseline				
GMT	37.5 (28.4–49.6)	5 (5–5)	32.9 (23.3–46.6)	5 (5–5)
Post primer dose				
GMT		6.9 (5.8–8.2)		5.9 (5.0–7.0)
GMT fold increase value		1.4 (1.2–1.6)		1.2 (1.0–1.4)
Post booster dose				
GMT	766.0 (528.1–1111.0)	53.2 (39.0–72.5)	71.6 (51.0–100.5)	6.6 (5.5–7.8)
GMT fold increase value	20.4 (14.8–28.1)	10.6 (7.8–14.5)	2.2 (1.4–3.3)	1.3 (1.1–1.6)

Data are GMT values (95% CI); PC-B: recovered individuals previously infected by COVID-19 receiving BNT162b2; PC-C: recovered individuals previously infected by COVID-19 receiving CoronaVac; CN-B: SARS-CoV-2 naïve individuals receiving BNT162b2; CN-C: SARS-CoV-2 naïve individuals receiving CoronaVac; post primer dose: serum collected on day 28 (CN-C) or day 21 (CN-B) post first dose of vaccine. Post booster dose: serum collected on day 56 (CN-C and CN-B) or on day 28 (PC-B and PC-C) post first dose of vaccine.

**Table 3 vaccines-09-01442-t003:** Immunogenicity of COVID-19 vaccine administered after or before 6 months.

Virus Strain	PC-B		*p* Value	PC-C		*p* Value
	Before 6 Months (*n* = 20)	Post 6 Months (*n* = 34)		Before 6 Months (*n* = 5)	Post 6 Months (*n* = 20)	
**Wild type**						
Baseline						
GMT	29.3 (19.6–43.7)	34.0 (24.2–47.7)	0.526	30.3 (17.6–52.2)	33.6 (23.9–47.2)	0.627
Post booster dose						
GMT	685.9 (463.0–1016.2)	738.2 (454.1–1200.1)	0.204	69.6 (21.6–224.1)	56.6 (42.4–75.4)	0.123
GMT fold increase value	23.4 (15.0–36.6)	21.7 (14.0–33.8)	0.613	2.3 (0.7–7.4)	1.7 (1.1–2.5)	0.155
**Delta variant**						
Baseline						
GMT	32.5 (22.2–47.7)	40.8 (27.8–59.8)	0.191	23.0 (13.8–38.2)	36.1 (23.9–54.3)	0.351
Post booster dose						
GMT	618.2 (409.4–933.4)	868.9 (507.1–1489.1)	0.066	60.6 (24.1–152.4)	74.6 (51.7–107.8)	0.910
GMT fold increase value	19.0 (12.7–28.5)	21.3 (13.5–33.5)	0.195	2.6 (1.3–5.3)	2.1 (1.2–3.4)	0.790

Data are GMT values (95% CI); PC-B: recovered individuals previously infected by COVID-19 receiving BNT162b2; PC-C: recovered individuals previously infected by COVID-19 receiving CoronaVac; post booster dose: serum collected on day 28 after receiving vaccine.

## Data Availability

The data used to support the findings of this study are included within the article.

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
