# Peer review of "Antibody Response of BNT162b2 and CoronaVac Platforms in Recovered Individuals Previously Infected by COVID-19 against SARS-CoV-2 Wild Type and Delta Variant"

_vaccines, 2021, doi:10.3390/vaccines9121442_

Round 1

Reviewer 1 Report

The manuscript describes the production of neutralizing antibodies against wild type and delta variants and in individuals vaccinated with Corona Vac and BNT162b2 vaccines. The study is clinically relevant, original only in terms of the cell assay to analyze neutralizing antibodies in time spam. There are limitations on the data of SARS CoV-2 infected patients (severity).

The amount of patients in different conditions is low which limits statistical analysis and conclusions. However, the data presented is relevant.

The discussion lacks proposals like the importance of the third boost after one year in any condition or the use of another vaccine to maintain memory response. It is also unclear if any of the patients vaccinated were exposed to Covid. 

Minor mistakes in grammar and text were observed.

Author Response

RE: A point-to-point response to Reviewer’s comments

“Antibody response of BNT162b2 and CoronaVac platforms in COVID-19 recovered individuals against SARS-CoV-2 wild type and delta variant” (vaccines-1474219) by Ruiqi Zhang, Ka-Wa Khong, Ka-Yi Leung, Danlei Liu, Yujing Fan, Lu Lu, Pui-Chun Chan, Linlei Chen, Kelvin Kai-Wang To, Honglin Chen, Kwok-Yung Yuen, Kwok-Hung Chan and

Ivan Fan-Ngai Hung.

We are grateful to the helpful comments given by referee. We have carefully revised the manuscript accordingly. A point-to-point response to reviewers’ comments is given as follow.

Comments:

1) The manuscript describes the production of neutralizing antibodies against wild type and delta variants and in individuals vaccinated with Corona Vac and BNT162b2 vaccines. The study is clinically relevant, original only in terms of the cell assay to analyze neutralizing antibodies in time spam. There are limitations on the data of SARS CoV-2 infected patients (severity).

Response: We thank the Reviewer for the comments. We have collected the information about the severity, and showed it in Table 1 and Results section.

Line 159-161: Among these patients, there were 9 severe and 70 mild COVID-19 cases (Table 1). No difference in percentage of severe COVID-19 cases between PC-B and PC-C groups (p=0.38).

2) The amount of patients in different conditions is low which limits statistical analysis and conclusions. However, the data presented is relevant.

Response: We thank the Reviewer for the comments. We agree that the sample size was not large, especially for patient receiving CoronaVac vaccine.

3) The discussion lacks proposals like the importance of the third boost after one year in any condition or the use of another vaccine to maintain memory response. It is also unclear if any of the patients vaccinated were exposed to Covid.

Response: We thank the Reviewer for the advice. One clinical study shows that BNT162b2 vaccine effectiveness against delta waned at 5 months after two doses vaccines 1. Thus, individuals should receive the third dose at 5 months after full vaccination. We have added the proposal on the third dose in the Discussion section.

Line 337-340: For the time of third dose booster in vaccinees, two clinical studies report a reduced effectiveness of BNT162b2 vaccine against SARS-CoV-2 at 6 months post-first dose [32,33]. Thus, vaccines should receive the third dose of COVID-19 vaccine 5 months after full vaccination.

As presented in Table 3, recruited healthy individuals recruited have no detectable neutralizing antibody against WT and DV at baseline, and the results were confirmed by iFlash RBD NAb assay (Shenzhen YHLO Biotech Co., Ltd., Shenzhen, China) which can detect anti-RBD antibody and anti-nucleocapsid protein antibodies. Furthermore, there are also no clinical record of SARS-CoV-2 infection. Therefore, we assume that they are SARS-CoV-2 naïve. Furthermore, the vaccination programme in the study started on May 1, 2021, and there were only 5 local cases (May 12, June 2 and 4, August 15, October 7, 2021) reported between May 1 and November 21, 2021 according to the government of HKSAR 2. The risk of these participants exposing to COVID-19 should be very low after vaccination.

4) Minor mistakes in grammar and text were observed.

Response: We thank the Reviewer for the comments. We have correct mistakes in grammar and text carefully.

Reference

  1. Tartof SY, Slezak JM, Fischer H, Hong V, Ackerson BK, Ranasinghe ON, Frankland TB, Ogun OA, Zamparo JM, Gray S et al: Effectiveness of mRNA BNT162b2 COVID-19 vaccine up to 6 months in a large integrated health system in the USA: a retrospective cohort study. Lancet 2021, 398(10309):1407-1416.
  2. Centre of health Protection, Department of Health, The Government of The Hong Kong Special Administration Region. https://chp-dashboard.geodata.gov.hk/covid-19/zh.html. Accessed on November 21, 2021.

Reviewer 2 Report

Zhang and colleagues presented a research article aimed at assessing the immune response rate of individuals after Covid-19 vaccination. For this purpose, the authors have evaluated the neutralization potentials of human antibodies obtained from individuals with a past SARS-CoV-2 infection who received a dose of two different types of vaccine (independently) and from SARS-CoV-2 naïve individuals who received two doses of vaccinations. The authors demonstrated a significant increase in the neutralizing effects of human antibodies after one and two doses of the vaccine while a decrement of neutralizing antibodies was observed after 1 year. Overall, the study is very interesting, however, there are some issues that the authors have to address before publication:
1) The results provided in the Abstract are described in a confusing manner. Please consider rewriting the abstract section in a clearer manner;
2) Please provide references supporting the following statements: “Different vaccine platforms have been adopted across the globe and in Hong Kong both mRNA vaccine and live-attenuated vaccine are available, currently there’s no evidence supporting the boosting effect of inactivated vaccine.”;
3) In the Introduction or Discussion sections the authors have to introduce the current and innovative methods for the diagnosis of SARS-CoV-2 and for the evaluation of the immune response of patients. The authors have to discuss differences in the methods for the detection of antibodies as well as for the evaluation of their neutralization effects. For this purpose, please see:
- PMID: 33846767
- PMID: 32311668
- PMID: 32607314
- PMID: 34563583
4) In Figure 2, have the authors shown the SD or SEM? Please, clarify;
5) Have the authors collected the socio-demographic and clinical-pathological data of the patients recruited in their study? It would be interesting to evaluate the immune response of patients according to the presence of comorbidities or other concomitant treatments. Please, clarify this issue;
6) Please consider shortening the Discussion section avoiding to add redundant information.

Author Response

RE: A point-to-point response to Reviewer’s comments

“Antibody response of BNT162b2 and CoronaVac platforms in COVID-19 recovered individuals against SARS-CoV-2 wild type and delta variant” (vaccines-1474219) by Ruiqi Zhang, Ka-Wa Khong, Ka-Yi Leung, Danlei Liu, Yujing Fan, Lu Lu, Pui-Chun Chan, Linlei Chen, Kelvin Kai-Wang To, Honglin Chen, Kwok-Yung Yuen, Kwok-Hung Chan and

Ivan Fan-Ngai Hung.

We are grateful to the helpful comments given by referee. We have carefully revised the manuscript accordingly. A point-to-point response to reviewers’ comments is given as follow.

Comments:

Zhang and colleagues presented a research article aimed at assessing the immune response rate of individuals after Covid-19 vaccination. For this purpose, the authors have evaluated the neutralization potentials of human antibodies obtained from individuals with a past SARS-CoV-2 infection who received a dose of two different types of vaccine (independently) and from SARS-CoV-2 naïve individuals who received two doses of vaccinations. The authors demonstrated a significant increase in the neutralizing effects of human antibodies after one and two doses of the vaccine while a decrement of neutralizing antibodies was observed after 1 year. Overall, the study is very interesting, however, there are some issues that the authors have to address before publication:

1) The results provided in the Abstract are described in a confusing manner. Please consider rewriting the abstract section in a clearer manner;

Response: We thank Reviewer for the comment. We have rewritten the abstract.

Line 21-38: Abstract: Vaccinating COVID-19 recovered patients with mRNA vaccines boosts their immune response against wild-type viruses (WT), we aimed to investigate whether vaccine platform and time of vaccination affect the immunogenicity against SARS-CoV-2 WT and delta variant (DV). Convalescent COVID-19 patients were recruited and received one booster dose of BNT162b2 (PC-B) or CoronaVac vaccines (PC-C), and SARS-CoV-2 naïve subjects were received two doses of BNT162b2 (CN-B) or CoronaVac vaccines (CN-C). The neutralizing antibody in sera against the WT and DV was determined with live virus neutralization assay (vMN). vMN geometric mean titre (GMT) against WT in COVID-19 recovered individuals reduced significantly from 60.0 (95% confidence interval (CI), 46.5-77.4) to 33.9 [95%CI, 26.3-43.7] at 6 months post-recovery. In PC-B group, BNT162b2 vaccine enhanced antibody response against WT and DV with 22.3-fold and 20.4-fold increase respectively. PC-C group also showed 1.8-fold and 2.2-fold increase for WT and DV respectively after receiving CoronaVac vaccine. There was a 10.6-fold increase of GMT in CN-B group and 1.3-fold increase in CN-C group against DV after full vaccination. In both PC-B and PC-C groups, there was no difference between GMT against WT and DV after vaccination. Subjects in CN-B and CN-C groups showed inferior GMT against DV compared to GMT against WT after vaccination. In the study, one booster shot effectively enhanced the pre-existing neu-tralizing activity against WT and DV in recovered subjects.

2) Please provide references supporting the following statements: “Different vaccine platforms have been adopted across the globe and in Hong Kong both mRNA vaccine and live-attenuated vaccine are available, currently there’s no evidence supporting the boosting effect of inactivated vaccine.”;

Response: We thank the Reviewer for the suggestion. We have amended the sentence and cited references are to supporting the statement.

Line 89-91: Different vaccine platforms have been adopted across the globe and in Hong Kong both BNT162b2 and CoronaVac vaccines are available [5,14]

3) In the Introduction or Discussion sections the authors have to introduce the current and innovative methods for the diagnosis of SARS-CoV-2 and for the evaluation of the immune response of patients. The authors have to discuss differences in the methods for the detection of antibodies as well as for the evaluation of their neutralization effects. For this purpose, please see:

- PMID: 33846767

- PMID: 32311668

- PMID: 32607314

- PMID: 34563583

Response: We thank Reviewer for the comment. We have added the information about diagnosis of SARS-CoV-2 in Introduction section, and discussion about method of immune response including antibodies to Discussion section. Moreover, these references (PMID: 33846767; PMID: 32311668; PMID: 32607314; PMID: 34563583) are also cited to support the information.

line 66-70: In response to COVID-19 pandemic, several diagnosis techniques for COVID-19 have been developed. There are three major methods for diagnosis: polymerase chain reaction (PCR)-based molecular test, rapid antigen and antibody test, and immune enzymatic serological test [4]. In addition, viral culture and imaging techniques are also ap-plied in COVID-19 diagnosis [4].

Line 341-352: In the laboratory, immune response induced by SARS-CoV-2 infection and vaccines can be evaluated by testing cytokines expression, antibody, and T lymphocytes [34-36]. For anti-SARS-CoV-2 antibody, it can be detected via vMN, ELISA, and immunofluo-rescence test (IF) [15,37]. ELISA and IF detect antibodies binding to SARS-COV-2. These antibodies contain neutralizing antibodies which protect host by inhibiting viral repli-cation, and non-neutralizing antibodies. Currently, surrogate neutralizing antibody (sNAb) test based on ELISA has been developed [34]. In this technique, RBD of SARS-CoV-2 spike protein was used to determine neutralizing antibodies which bind to RBD. Compared with sNAb test, vMN used in the study can detect neutralizing anti-bodies including not only anti-RBD antibody but also anti-other epitope neutralizing antibody, for instance, antibodies against N-terminal domain (NTD) of spike [15,38]. Thus, results from vMN can evaluate the neutralizing activity induced by vaccine and infection comprehensively.

4) In Figure 2, have the authors shown the SD or SEM? Please, clarify;

Response: We thank the Reviewer for the comments. 95% CI was shown in the Figure 2. And we have added it to figure legend.

Line 174-175: Each dot represents an individual serum sample, and error bar represents the 95% confidential interval (CI).

5) Have the authors collected the socio-demographic and clinical-pathological data of the patients recruited in their study? It would be interesting to evaluate the immune response of patients according to the presence of comorbidities or other concomitant treatments. Please, clarify this issue;

Response: We thank the Reviewer for the comments. We agree that the socio-demographic and clinical-pathological information of patients would be useful in the evaluation of their immune response. The data of comorbidities and severity of disease have been added to the manuscript and presented in Table 1.

6) Please consider shortening the Discussion section avoiding to add redundant information.

Response: We thank the Reviewer for the comment. We have shortened Discussion section by deleting the sentences as follow:

Line 256: using Next Generation Sequencing (NGS),

Line 257-258: 142 days after first symptomatic infection
Line 257-259: of the serum antibody profile of this patient

Line 291: After demonstrating the efficacy of mRNA vaccine in COVID-19 patients

Line 292-295: using vaccines with different platforms as different vaccine platforms are adopted across the globe. In Hong Kong, both mRNA vaccine (BNT162b2) and inactivated vaccine (CoronaVac) are available.

Line 310-311: neutralizing antibody titre

Line 362-366: In addition, our study demonstrated that for SARS-CoV-2 naïve individuals, two doses of BNT162b2 protects against DV but for individuals who received two doses of CoronaVac, protection against DV is minimal, therefore, an additional mRNA booster dose should be consider to enhance their neutralizing antibody response against SARS-CoV-2 variants

Reviewer 3 Report

The authors aimed to evaluate the persistence of antibody level in COVID-19 recovered patients, the boosting effect on antibody level of these patients with different vaccine platform, and whether such vaccination protects them from DV and the most appropriate timing of administrating the vaccine.

The study covers some issues that have been overlooked in other similar topics. The structure of the manuscript appears adequate and well divided in the sub-paragraphs. Moreover, the study is easy to follow.

Introduction section: Will be useful to the reader to add some interesting recent literature about the updates against molecular mechanisms related to  SARS-CoV-2 outbreak and related tools to counteract the same (please see and briefly discuss:  PMID: 33024749).

Conclusion Section: This paragraph is missing. Please add it.

Author Response

RE: A point-to-point response to Reviewer’s comments

“Antibody response of BNT162b2 and CoronaVac platforms in COVID-19 recovered individuals against SARS-CoV-2 wild type and delta variant” (vaccines-1474219) by Ruiqi Zhang, Ka-Wa Khong, Ka-Yi Leung, Danlei Liu, Yujing Fan, Lu Lu, Pui-Chun Chan, Linlei Chen, Kelvin Kai-Wang To, Honglin Chen, Kwok-Yung Yuen, Kwok-Hung Chan and

Ivan Fan-Ngai Hung.

We are grateful to the helpful comments given by referee. We have carefully revised the manuscript accordingly. A point-to-point response to reviewers’ comments is given as follow.

Comments

1) The authors aimed to evaluate the persistence of antibody level in COVID-19 recovered patients, the boosting effect on antibody level of these patients with different vaccine platform, and whether such vaccination protects them from DV and the most appropriate timing of administrating the vaccine.

Response: We thank the Reviewer for the comments.

2) The study covers some issues that have been overlooked in other similar topics. The structure of the manuscript appears adequate and well divided in the sub-paragraphs. Moreover, the study is easy to follow.

Response: We thank the Reviewer for the comments.

3) Introduction section: Will be useful to the reader to add some interesting recent literature about the updates against molecular mechanisms related to SARS-CoV-2 outbreak and related tools to counteract the same (please see and briefly discuss:  PMID: 33024749).

Response: We thank the Reviewer for the suggestion. We add some information about SARS-CoV-2 outbreak and transmission, and method to control it. We also cite the reference (PMID: 33024749) to support it.

Line 60-66: COVID-19 is caused by severe acute respiratory syndrome coronavirus 2 (SARS-CoV-2), which is a type of enveloped and positive-stranded RNA virus [1]. The receptor-binding domain (RBD) of spike protein, which is the membrane protein of SARS-CoV-2, can bind to angiotensin-converting enzyme 2 (ACE2) on surface of host cells [2]. Virus then en-ters the cell and completes viral replication. New virions can transmit among the human population through aerosol, droplet, and fomite [3]

Line 70-76: Although these techniques can help us screen the new COVID-19 cases quickly and ef-ficiently to tackle the pandemic, vaccine is still the major method to control COVID-19. COVID-19 vaccines approved for clinical use include mRNA vaccine (BNT162b2), viral vector -based vaccine (ChAdOx1), inactivated virus vaccine (CoronaVac), and subunit protein vaccine (NVX-CoV2373) [5]. Vaccines can induce immune response against SARS-CoV-2 to protect host which can also be elicited by virus infection [6].

4) Conclusion Section: This paragraph is missing. Please add it.

Response: We thank the Reviewer for the advice. We added the conclusion section into the manuscript.

Line 368-376: 5. Conclusion

Our study found that one BNT162b2 or CoronaVac vaccine booster could enhance the pre-existing antibody response in recovered patients against WT and DV effective-ly. Furthermore, our study demonstrated that for SARS-CoV-2 naïve individuals, two doses of BNT162b2 protects against DV but for individuals who received two doses of CoronaVac, protection against DV is minimal, therefore, an additional mRNA booster dose should be considered to enhance their neutralizing antibody response against SARS-CoV-2 variants. This study could provide useful information about the time of vaccination in COVID-19 recovered individuals.

Round 2

Reviewer 2 Report

The authors have well-addressed all of my previous comments. The manuscript was significantly improved and can be accepted for publication after the editorial check.